# Quality of Life and Its Associated Factors among Patients with Psoriasis in a Semi-Urban Northeast Malaysia

**DOI:** 10.3390/ijerph191811578

**Published:** 2022-09-14

**Authors:** Mohd Shahriman Ahmad Fuat, Zainab Mat Yudin, Juliawati Muhammad, Faridah Mohd Zin

**Affiliations:** 1Department of Family Medicine, School of Medical Science, Universiti Sains Malaysia, Kota Bharu 16150, Kelantan, Malaysia; 2School of Dental Science, Universiti Sains Malaysia, Kota Bharu 16150, Kelantan, Malaysia; 3MSU Medical Centre, Jalan Boling Padang 13/64, Seksyen 13, Shah Alam 40100, Selangor, Malaysia

**Keywords:** psoriasis, depression, quality of life, chronic disease

## Abstract

Psoriasis is a chronic relapsing dermatological disorder that significantly affects the patients’ psychosocial well-being and quality of life (QOL). This study aimed to determine the proportion of severely impaired QOL, the factors associated with severely impaired QOL, and its correlation with depression among semi-urban populations on the Northeast Coast of the Peninsular Malaysia. A cross-sectional study was conducted among 257 patients with psoriasis at the Dermatology Clinic of Hospital Sultanah Bahiyah via a self-administered questionnaire that included sociodemographic profiles, the validated Malay version of the Dermatology Life Quality Index (DLQI), and the Malay version of the Beck depression scale. The data were analysed using logistic and linear regression models. About 20.5% of the patients had severely impaired QOL quality of life, while 79.5% of the patients had non-impaired QOL. Multiple logistic regression analysis showed that the psoriatic severity [Adjusted OR = 1.91, 95% CI: 1.76, 9.93; *p* < 0.001] and exposed area [Adjusted OR 2.93, 95% CI: 0.38, 2.29; *p* = 0.050] had a significant association with severely impaired QOL. Among the patients, 18.7% had a positive result in the screening for depression, which revealed a significant association between QOL and depression scores [r = 0.47, 95% CI: 0.35, 0.56, *p* < 0.001]. Psoriasis can impair QOL and have a relation with mental health. Regular screening for depression among patients with psoriasis is a beneficial strategy for the early detection of depression, especially in semi-urban areas.

## 1. Introduction

Psoriasis is a relatively common inflammatory, proliferative skin disease that affects about 1 to 3% of the world’s population [1,2]. It is a complex, multifaceted skin disorder that can significantly affect the patient’s QOL, influencing daily, social activities, and other aspects of life [3]. Clinicians assess the severity of psoriasis on the basis of clinical signs such as the extent of body surface area (BSA) involvement and the degree of scaliness, redness, and thickness using the Psoriasis Area Severity Index (PASI) [2]. Its pathogenesis and progression are related to genetic factors, emotional stress, drugs, infections, and physical trauma [4]. Psoriasis has been reported to severely interfere with all aspects of a patient’s life, unlike lives compared to patients with other skin disorder [5]. The degree of deterioration in the QOL of patients was found to be statistically significant in terms of all four dimensions, namely symptoms, emotions, life, and social functions [4].

Many studies have reported the factors associated with the QOL of patients with psoriasis. Many associated factors can contribute to severely impaired QOL such as sociodemographic and psoriasis factors [1,6,7]. Malaysia is a multi-ethnic country, and many studies have been conducted to determine the sociodemographic that can impair the QOL of patients with psoriasis [6,8,9,10,11]. This study aimed to examine the factors associated with QOL and correlate depression and QOL in patients with psoriasis in semi-urban areas.

## 2. Materials and Methods

This cross-sectional quantitative study was conducted among 257 patients with psoriasis from January to June 2019. We used convenience sampling involving all patients aged 18 years and above.

### 2.1. Setting

This study was conducted at the Dermatology Clinic of Hospital Sultanah Bahiyah, Kedah, a state located on the Northeast Coast, Malaysia. It is a tertiary hospital in a semi-urban area and receives referrals from district hospitals, health clinics, and private clinics nearby.

### 2.2. Participants

All patients registered in the psoriatic registry system in the hospital were included in this study. Participants who met the inclusion criteria, whereas patients with a concomitant psychotic psychiatric disorder and established diagnosis of depression, were excluded.

### 2.3. Research Tools

This study used a self-administered questionnaire that included the sociodemographic profile, the validated Malay version of Dermatology Life Quality Index (DLQI), and the Beck Depression Inventory-Malay (BDI-Malay). A sociodemographic questionnaire included background data such as age, race, sex, marital status, educational status, employment status, and household income. The medical records were reviewed for the types of psoriasis, age of onset, joint involvement, severity, and the affected anatomical location of the disease. The types of psoriasis were categorised into plaque and non-plaque. The disease’s age of onset was categorised as either <30 or 30 years old and above. The joint involvement was classified into with or without joint involvement. The severity of the disease was classified as either non-severe (PASI score <10) or severe (PASI score ≥10). The DLQI and BDI-Malay are described in Table 1.

### 2.4. Data Analyses

Data entry and analysis were conducted using IBM SPSS Statistics ver. 26.0 (IBM Corp., Armonk, NY, USA). We used descriptive statistics including the means and standard deviations (SD) to describe the sociodemographic, clinical, and QOL characteristics. Simple and multiple logistic regressions were carried out to identify the associated psoriatic and sociodemographic factors for severely impaired QOL. Simple logistic regression for the initial model was performed. The variables that met the initial screening criterion of *p* < 0.25 were then regressed using multiple logistic regression with forward, backward, and stepwise methods.

## 3. Results

### 3.1. Characteristics of the Participants

The mean (SD) age of all the 257 patients who were enrolled and participated in this study was 46.6 (16.7) years old. Those of patients belonging to the severely impaired and non-impaired QOL groups were 38.8 (14.2) and 48.6 (16.7) years old, respectively. The socio-demographic characteristics of the patients and the clinical features of psoriasis are shown in Table 2 and Table 3, respectively.

### 3.2. Associated Factors by Logistic Regression

There were significant associations between the severity and anatomy psoriasis in patients with severely impaired QOL (Table 4 and Table 5). Patients with severe psoriasis had a 1.91-fold higher probability of having severely impaired QOL than those with non-severe psoriasis. Those who had exposed and non-exposed psoriatic lesions had 2.93-fold and 1.41-fold higher probabilities of having severely impaired QOL, respectively, than those who had both lesions.

### 3.3. Association between Depression Scores and QOL

There was a positive correlation between the QOL and depression scores with a beta coefficient value of 1.47 with 95% CI (0.35, 0.56) and a *p*-value < 0.001 (Table 6).

## 4. Discussion

### 4.1. Prevalence of Severely Impaired QOL

Our study yielded a slightly lower prevalence rate of severely impaired QOL for this disease than that of the general population in Malaysia, which is about 33%, as reported in 2014, 2015, and 2018 [1,16]. The reports were multicentric national registry surveys in all Malaysian hospitals that provided psoriasis treatment and used a set of questionnaires similar to our study. Even though a similar questionnaire was used, our study’s prevalence rate of severely impaired QOL was also lower than those of the Asia Pacific (33.6%) and global (36.5%) populations [10,17].

This study was conducted in a semi-urban area of Malaysia. Most patients were Malay, married, and employed with good sociodemographic support. Thus, from the patients’ cultural and sociodemographic backgrounds, the level of acceptance was high, and the patients with psoriasis coped well with the stress associated with the disease [18]. A study conducted in Malaysia found that impaired QOL in chronic disease depends on the cultural perception, study population such as rural/urban population, and sociodemographic factors of the patients [8,18]. One study conducted by Chan et al. in 2015 showed that people in rural areas were more likely to have severely impaired QOL than those in semi-urban areas. This may be related to limited access to health care services and a lack of financial means to pay for such services [8].

Another study conducted at urban centre hospitals by Tang et al. in 2013 showed that the prevalence of severely impaired QOL among patients with psoriasis was higher than that in our study, which was about 46% [10]. The higher prevalence of severely impaired QOL in their research was most likely attributable to other factors such as urban areas, which have many stressful surrounding factors that can affect the QOL. Our study was conducted in a semi-urban area, where the economic and surrounding factors are different from those in urban areas [10].

One study conducted in China by Ching et al. in 2016 showed that the prevalence of severely impaired QOL among patients with psoriasis was 48.1%, which was higher than that of the global population [19]. There were differences in the social factors, surroundings, and medical services; the population was more concerned with physical appearance and personality in daily activities. Thus, it might make them feel embarrassed, flawed, and stigmatised in the community [19,20].

### 4.2. Severity of Psoriasis

Our results showed that disease severity and the lesion’s anatomical location are significantly associated with severely impaired QOL among patients with psoriasis. In this study, the severity of psoriasis was associated with severely impaired QOL, and our findings were similar to those of previous Malaysian [6,10], Asia Pacific, Asia Pacific, and global studies [4]. In Malaysia, the 2014 National Psoriasis Registry showed an association between disease severity and significantly impaired QOL (*p* < 0.001). The registry recorded all the psoriasis patient data from 2012 to 2018, consisting of 11,622 registered patients. About 10% of the patients had a PASI score of ≥10, which was significantly related to severely impaired QOL [1,16]. They also reported that most (about 53%) of the psoriasis patients in Malaysia had mild to moderate BSA involvement, and severe psoriasis with >10% BSA involvement occurred in 16.1% [16].

In 2013, Min moon et al. reported that the most important factor affecting QOL was the clinical severity of psoriasis, as indicated by the PASI score. Patients with higher PASI scores had a lower QOL, as evidenced by statistically significantly higher DLQI scores. Patients with higher PASI scores had a lower QOL, as evidenced by statistically significantly higher DLQI scores [10]. Their results were comparable to those in our study as their study was also conducted in a dermatology clinic in a semi-urban area in Malaysia. Most patients with psoriasis were severely affected by the symptoms of itch and pain caused by the disease and felt embarrassed.

A study by Chen et al. in China conducted in 2016 summarised a similar finding to our study, in which a higher severity was confirmed as the most crucial determinant for a lower QOL. People with severe disease had a 7.3-fold significantly higher risk of having a worse QOL [19]. However, in 2015, Nazemei et al. reported in Iran that the psychological vulnerability score (SCL90), illness perception score (IPQ-R), and coping strategies score (CISS) had no significant relationship with the PASI score [21]. Their questionnaire was different from our DLQI questionnaire, which specifically considers the psychological impairment of the patient. Studies in Spain and the USA showed similar findings regarding the significant association between psoriatic severity and severely impaired QOL. It was concluded that although the severity of psoriasis is significant, there are other critical aspects of psoriasis care including body pain and mental well-being [22,23,24].

### 4.3. Anatomy of the Lesions

Our study showed a significant association between the anatomy of lesions and severely impaired QOL. Our findings were similar to those of other Malaysian, Asia Pacific, and global studies [6,9,19]. Nyunt et al. (2013) at University Malaya Medical Centre showed that about 58.8% had significantly impaired QOL in patients with exposed lesions. About 46.8% had suffered exposed area involvement in patients with psoriasis [6]. This result was almost similar compared to our study, where approximately 46.2% suffered exposed area involvement, and 28.8% sustained both area involvement. The cosmetic disfigurement may make a person with psoriasis feel embarrassed, flawed, and stigmatised, disrupting personal relationships, work-related, and leisure activities [25].

Most previous studies have evaluated the exposed and non-exposed lesion areas but did not classify both involvements of the site as one category [6,17,26]. Our study revealed that patients who showed only exposed area involvement had a more impaired QOL than those who showed both exposed and non-exposed lesion involvements.

In 2015, a study by Yang et al. of 200 patients in a dermatology clinic in Taiwan, showed a significant association between the exposed lesion (hand) and severely impaired QOL [17]. They showed that patients with psoriasis involving their hands were more depressed (59.1%) and their life, social, and occupational functions were more severely affected. Most patients with psoriasis involving their hands worried more about losing their job than patients with non-exposed lesions [17].

This study could not establish any association between the sociodemographic data and severely impaired QOL among patients with psoriasis. Likewise, studies in Malaysia showed inconclusive results regarding sociodemographic factors related to QOL among the Malaysian population [7,9,27]. A similar finding can be found worldwide: there was no association between demographic characteristics and severely impaired QOL [17,28].

### 4.4. Prevalence of Depression among Patients with Severely Impaired QOL

In this study, we found that the prevalence of depression among patients with severely impaired QOL was almost 19%, with a mean score of 6.72 (SD 4.412). A recent study at a similar site using the Hospital Anxiety and Depression Scale (HADS) to assess depression found about 8.5% of patients had depression [29]. Generally, the proportion of psoriasis patients with depression symptoms ranged from 9% to 55%, likely due to differences in the screening methods and study populations [4]. A meta-analysis conducted in 2014 found that the use of different questionnaires resulted in differences in the prevalence of depression. The study showed that utilising BDI resulted in a higher prevalence of patients with depression symptoms at 26% [30]. Compared with other questionnaires used to assess depression symptoms, BDI is more specific and focused on diagnosing depression [14].

The prevalence rate of depression was lower in this study than the global prevalence rate in patients with psoriasis, which is 30% to 65% [17,31]. A study by Maria et al. conducted in Italy in 2006 showed that the prevalence of depression among patients with psoriasis, especially among female patients, was about 62% [31]. Cultural influences may play a part in the relatively low prevalence of patients with depression in our study. In general, lifetime rates of depression are significantly much higher in the West than in Asian countries. This difference has been theorised to be due to the better ways of managing negative emotions in the Asian culture, which prevents symptoms from escalating into a mood disorder [18].

### 4.5. Association between DLQI Score and Depression Score

In this study, we showed a positive correlation between the DLQI score and depression score, with the beta coefficient value of 1.47 with 95% CI (0.35,0.56) and the *p*-value of <0.001. The more severe the QOL, the higher the risk of the patient developing depression. This finding is a similar result to those of other studies around the world. Ghajarzadeh et al. found that depressed individuals tended to have higher DLQI scores (*p* = 0.00) [15]. They discovered that maladaptive coping responses, body image problems, and feelings of shame, stress, and worry had been suggested to cause depression in psoriasis patients.

In 2017, Mahmutovic et al. found that the degree of depression and the QOL of persons with psoriasis were negatively correlated. As the values of the QOL domains become poorer, the respondents’ degree of depression grows. Their study focused on two domains of QOL, namely physical and psychological health, which correlate with depression symptoms. Both domains showed a significant relation with a *p*-value of <0.001 [32]. In comparison, other studies in Malaysia showed similar findings to ours, revealing a significant correlation between the QOL score and depression score [1,10].

## 5. Conclusions

The QOL of patients with psoriasis is affected by multiple factors. They also demonstrated having more depression symptoms, which correlated with the QOL. Patients with more severe psoriatic characteristics require further evaluation of their mental health. Screening for depression symptoms among psoriasis patients is necessary. The limitation of our study is that it only evaluated patients at a tertiary centre in which patients suffering from less severe psoriasis might be underrepresented. Assessing this issue in a broader population setting can add more lines of evidence in future studies.

## Figures and Tables

**Table 1 ijerph-19-11578-t001:** The description of DLQI and BDI-Malay.

Questionnaire	Questions/Items	Scoring	Cronbach Alpha
DLQI-Malay [6,12]	10 questions; symptoms and feelings (Q1 & Q2), daily activities (Q3 & Q4), leisure (Q5&Q6), work and school (Q7), personal relationships (Q8 & Q9), and treatment (Q10).	It being scored from 0 to 3 (a little, a lot, or very much). Minimum = 0 and Maximum = 30. If >10 (severely impaired) and ≤10 (non-impaired)	0.844
BDI-Malay [13,14,15]	20 items, a self-report rating inventory on attitudes and symptoms of depression over the past week.	Scores ≥ 10 (depressed) Scores < 10 (normal)	0.91

**Table 2 ijerph-19-11578-t002:** The sociodemographic characteristics of patients (n = 257).

Variables	Total n = 257 (100%)	Non-Impaired QOL (n = 205) 79.5%	Severe Impaired QOL (n = 52) 20.5%
Age (year) ^a^ Mean (SD): 46.6 (16.7) ^a^		48.6 (16.7) ^a^	38.8 (14.2) ^a^
Sex:			
Male	135 (52.5)	107 (52.2)	28 (53.8)
Female	122 (47.5)	98 (47.8)	24 (46.2)
Race			
Malay	227 (88.3)	182 (88.8)	45 (86.5)
Chinese	16 (6.2)	12 (5.9)	4 (7.7)
Indian	12 (4.7)	11 (5.4)	1 (1.9)
Others	2 (0.8)	0	2 (3.8)
Marital status			
Married	186 (88.3)	154 (75.1)	32 (61.5)
Not married	64 (24.9)	45 (22)	19 (36.5)
Divorced	7 (2.7)	6 (2.9)	1 (1.9)
Occupation			
Employed	160 (62.3)	123 (60.0)	37 (71.2)
Non employed	97 (37.7)	82 (40.0)	15 (28.8)
Education level			
Primary	35 (13.6)	32 (15.6)	3 (5.8)
Secondary	142 (55.3)	117 (57.1)	25 (48.1)
Tertiary/University	80 (31.1)	56 (27.3)	24 (46.2)
Family with psoriasis			
Yes	78 (30.4)	58 (28.3)	20 (38.5)
No	179 (69.6)	147 (71.7)	32 (61.5)
Income (RM) ^b^			
≤1000	109 (42.4)	92 (42.4)	17 (42.5)
>1001–3000	101 (39.3)	85 (39.2)	16 (40.0)
>3001	47 (18.3)	40 (18.4)	40 (18.4)

^a^ mean (SD), ^b^ median (IQR).

**Table 3 ijerph-19-11578-t003:** The clinical characteristic of patients (n = 257).

Variables	Total n = 257 (100%)	Non-Impaired QOL (n = 205) 79.5%	Severe Impaired QOL (n = 52) 20.5%
Type of psoriasis			
Plaque	246 (95.7)	207 (95.4)	39 (97.5)
Non plaque	11 (4.3)	10 (4.6)	1 (2.5)
Age of onset of psoriasis			
≤30 years old	116 (45.1)	86 (42.0)	30 (57.7)
30> years old	141 (54.9)	119 (58.0)	22 (42.3)
Psoriatic severity			
Severe	44 (17.1)	27 (13.2)	17 (32.7)
Non severe	213 (82.9)	178 (86.8)	35 (67.3)
Location of psoriasis			
Exposed	89 (34.6)	65 (31.7)	24 (46.2)
Non exposed	116 (45.1)	103 (50.2)	13 (25.0)
Both	52 (20.2)	37 (18.0)	15 (28.8)
Joint/nail involvement			
Yes	103 (40.1)	75 (36.6)	28 (53.8)
No	154 (59.9)	130 (63.4)	24 (46.2)

**Table 4 ijerph-19-11578-t004:** The associated factors for quality of life using simple logistic regression.

Variables	Crude OR ^a^	(95% CI ^b^)	Wald Stat ^c^	*p* Value
Age	0.02	(0.006, 6.664)	0.809	0.368
Sex				
Male	1.00			
Female	0.788	(0.399, 1.558)	0.468	0.494
Race				0.648
Malay	1.960	(0.596, 6.443)	1.227	0.268
Chinese	0.534	(0.067, 4.278)	0.349	0.555
Indian	1.00			0.999
Marital status				0.178
Married	1.971	(0.964, 4.032)	3.457	0.063
Not married/divorced	1.00			
Income (RM)				
<1000				0.986
1001–3000	1.019	(0.484, 2.143)	0.002	0.961
3001–6000	1.015	(0.368, 2.797)	0.001	0.977
>6000	0.676	(0.079, 5.763)	0.128	0.721
Occupation				
Employed/schooling	0.578	(0.274, 1.218)	2.080	0.149
Non employed	1.00			
Education Level				0.016
Primary	2.395	(0.529, 10.847)	1.285	0.257
Secondary	5.500	(1.210, 25.005)	4.868	0.027
University/above	1.00			
Family history of psoriasis				
Yes	0.682	(0.337, 1.379)	1.137	0.286
No				
Types				0.961
Plaque	0.000	(0.000, 0.000)	0.000	0.999
Guttate	0.000	(0.235, 29.986)	0.000	1.000
Erythrodermic	2.654	(0.000)	0.622	0.430
Pustular	0.000		0.000	1.000
Age of diagnosis				
<30 years old	0.554	(0.280, 1.097)	2.874	0.090
>30 years old	0.247		35.992	
Severity				
Yes (PASI ≥ 10)	0.257	(0.121, 0.544)	12.597	0.000
No (PASI < 10)	0.517		4.297	0.038
Joint involvement				
YES	0.382	(0.191, 0.761)	7.479	0.006
NO	0.304		26.128	0.000
Anatomy of psoriasis				0.008
Exposed	0.332	(0.141, 0.780)	6.403	0.011
Non exposed	1.315	(0.583, 2.965)	0.435	0.510
Both	0.254		27.042	0.000

^a^ Adjusted odds ratio, ^b^ Confidence interval, ^c^ Wald statistic. Note: No significant interaction; no multicollinearity; model assumptions met.

**Table 5 ijerph-19-11578-t005:** The associated factors for severely impaired quality of life using multiple logistic regression.

Variables	Adj. OR ^a^	95% CI ^b^	Wald Stat ^c^	*p* Value
Severity				
Severe	1.91	(1.76, 9.93)	10.50	0.001
Non-severe	1.00			
Anatomy of psoriasis				
Exposed	2.93	(0.38, 2.29)	3.56	0.050
Non-exposed	1.41	(0.15, 1.15)	2.65	0.040
Both	1.00			

^a^ Adjusted odds ratio, ^b^ Confidence interval, ^c^ Wald statistic. Note: No significant interaction; no multicollinearity; model assumptions met.

**Table 6 ijerph-19-11578-t006:** The association between the depression score and quality of life score among patients with psoriasis.

Variable	n	Adj b ^d^ (95% CI ^a^)	t Statistic	*p*-Value ^b^
QOL score	257	0.47 (0.35, 0.56)	8.604	0.001

^a^ Confidence interval. ^b^ Multiple linear regression (model assumptions met). Adjusted for psoriatic severity and anatomical location of psoriasis. ^d^ Adjusted regression coefficient.

## Data Availability

The data are contained within the article.

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
