# Peer review of "Quality of Life and Its Associated Factors among Patients with Psoriasis in a Semi-Urban Northeast Malaysia"

_ijerph, 2022, doi:10.3390/ijerph191811578_

Round 1

Reviewer 1 Report

The authors presented an interesting paper on quality of life (QOL) in psoriasis (Ps) patients. They evaluated the influence of disease-related and socioeconomic factors on the QOL of patients with Ps. Severity and anatomy of Ps had an impact on QOL. In Table 1, the authors presented sociodemographic and clinical characteristics of patients. In my opinion, an important factor that should have been included in Table 1 was the treatment used or lack of treatment. We know that the use of biologic treatment such as TNF alpha blockers, IL-17 or IL-23 blockers significantly reduces the activity and severity of Ps. Therefore, we do not know whether patients who had severe Ps were effectively treated. The authors conclude that regular screening for depression will be beneficial in Ps patients. I agree, but perhaps the lack of treatment was the cause of severe Ps and this continued to be a factor in exacerbating depression. I believe that these data should be supplemented before the paper is published.

Author Response

Dear Reviewer 1,

We are very delighted to receive such valuable comments, but unfortunately, it is our study limitation that we didn't include the treatment as part of the variable.

The responses to the comments are attached below:

Reviewer 2 Report

Dear Authors, 

I read with interest your article. It is a well-written article. However, the only novel contribution it should make, which is related to the specific geographical area where the research is carried out, does not seem to be of great importance in the light of the results described: The multi-culturality of the area does not seem to have an impact on the quality of life of patients with psoriasis, and data already widely described in the scientific literature, such as the severity or location of lesions, do. On the other hand, the relationship between psoriasis and depression has also been well documented on several occasions in the scientific literature.

1) In the abstract, line 21, the confidence interval should be revised as I think it is not correct. 

2) Were specific criteria followed for diagnosing joint involvement? Such as CASPAR criteria? If validated criteria for psoriatic arthritis were not used, explain how the diagnosis of joint involvement was arrived at (symptomatology reported by the patient? Diagnosis based on previous medical history?)

3) Univariate analysis should be performed when comparing "Non-impaired QOL" and "Severe impaired QOL". A new table should be added including p values.

4) Age and sex-matched controls would be of great interest to evaluate how the depressive symptoms are more frequent among patients with psoriasis. 

5) The fact that the study has been performed in a tertiary hospital should be added as a limitation, as it could mean that patients suffering from less severe psoriasis are underrepresented. 

Author Response

Dear Reviewer,

Thank you for the meaningful comments. They help us to improve our manuscript. Here we attached the responses to the comments.

Reviewer 3 Report

Dear Authors,

thank you for submitting your interesting paper. The topic seems to be attractive and interesting, however there are several significant minors of the research.

Introduction even though it is comprehensive, it's unquestionably too long. I think you should focus more directly on the topic. Also there are some repetitions in the introduction.

Furthermore material and methods should be shortened. In my opinion there is a high need to put those information into the table rather than in the text.  Draw attention on the data analysis subsection, it's also comprehensive but also too long. 

Results of the study are interesting and well grouped. However the tables could be planned better (divided). The table 2. is totally unclear for the reader. 

195 "About 10% of patients had a PASI score of ≥ and.." There is a mistake in the sentence, you did not mention the direct PASI score.

Conclusions are clear and understandable. 

Author Response

Dear Reviewer,

Thank you very comments. We are very grateful to have such comments and improve our manuscript better.

The responses are attached below:

Round 2

Reviewer 2 Report

Dear Authors,

The suggested changes have been improved.

Author Response

Thank you for the feedback.

We have sent the manuscript for scientific English proofreading, and changes have been made accordingly.

Reviewer 3 Report

I approve corrections made by authors

Author Response

Thank you for the approval of corrections.